# Overcoming Poor Transgene Expression in the Wild-Type *Chlamydomonas* Chloroplast: Creation of Highly Mosquitocidal Strains of *Chlamydomonas reinhardtii* [note 1]

**DOI:** 10.3390/microorganisms10061087

**Published:** 2022-05-25

**Authors:** Obed W. Odom, Seongjoon Kang, Caleb Ferguson, Carrie Chen, David L. Herrin

**Affiliations:** 1Department of Molecular Biosciences, University of Texas at Austin, Austin, TX 78712, USA; owodom@utexas.edu (O.W.O.); cmferguson3@gmail.com (C.F.); carrie.chen@utexas.edu (C.C.); 2Pond Life Technologies LLC, Cedar Park, TX 78613, USA; joonk@utexas.edu

**Keywords:** *Aedes aegypti*, biolarvacide, *Chlamydomonas reinhardtii*, *cry11Aa*, chloroplast engineering, green microalga

## Abstract

High-level expression of transgenes in the chloroplast of wild-type *Chlamydomonas reinhardtii* (*C. reinhardtii*) remains challenging for many genes (e.g., the *cry* toxin genes from *Bacillus thuringiensis israelensis*). The bottleneck is presumed to be post-transcriptional and mediated by the 5′ element and the coding region. Using 5′ elements from highly expressed photosynthesis genes such as *atpA* did not improve the outcome with *cry11A* regardless of the promoter. However, when we employed the 5′ UTR from **mature** *rps4* mRNA with clean fusions to promoters, production of the rCry11A protein became largely promoter-dependent. The best results were obtained with the native 16S *rrn* promoter (−91 to −1). When it was fused to the mature 5′ *rps4* UTR, rCry11A protein levels were ~50% higher than was obtained with the inducible system, or ~0.6% of total protein. This level was sufficient to visualize the 73-kDa rCry11A protein on Coomassie-stained gels of total algal protein. In addition, analysis of the expression of these transgenes by RT-PCR indicated that RNA levels roughly correlated with protein production. Live cell bioassays using the best strains as food for 3rd instar *Aedes aegypti* larvae showed that most larvae were killed even when the cell concentration was as low as 2 × 10^4^ cells/mL. Finally, the results indicate that these highly toxic strains are also quite stable, and thus represent a key milestone in using *C. reinhardtii* for mosquito control.

## 1. Introduction

High-level expression of transgenes in the chloroplast of wild-type *C. reinhardtii* remains a challenge [1,2,3,4,5,6]. The problem is an expression bottleneck that seems to occur at the post-transcriptional level and involves the 5′ element(s) of the transgene, and the coding region. This limitation arises because of the need to use a 5′ element on the transgene that is homologous to a 5′ element from an endogenous chloroplast gene. Completely heterologous 5′ elements that are effective at promoting translation, processing, and stability of mRNAs in the *C. reinhardtii* chloroplast have not been identified [7,8].

Most of the 5′ elements used to date are from highly expressed genes for photosynthesis [7,8,9]. These elements seem to be the target of at least one, and sometimes more, specific factor(s) that promotes the processing, stability, and/or translation of the mRNA [10,11]. Thus, one cause of poor transgene expression seems to be that the transgene mRNA does not compete well with the endogenous mRNA for the factor(s) that interacts with their shared 5′ elements. Why the transgenic mRNA should be a poorly competitive substrate for the specific 5′ factors is not clear, but there is evidence that the transgene’s coding region is involved, at least in some cases [4, our unpublished results]. It may be that the coding region induces alternate structures with the 5′ element that inhibit binding of the stabilizing/activating factors.

Another mechanism of poor transgene expression in a wild-type background appears to involve the protein encoded by the endogenous gene that was the source of the homologous 5′ element [5,6]. This mechanism resembles feedback repression, which may be a normal mechanism used to limit expression of certain chloroplast genes (although repression appears to be more severe for the transgene than the 5′-homologous endogenous gene).

For our purposes, we want a robust wild-type strain of *C. reinhardtii* as the host for the transgenes, because the algae are to be used for mosquito control in a variety of aquatic environments. Our approach is to develop strains of *C. reinhardtii* that express one or more genes in the chloroplast that are based on the toxin from *Bacillus thuringiensis* subsp. *israelensis* (Bti), which specifically kills mosquito and black fly larvae, and does so with little effect on the environment [12]. The Bti toxin is a crystal-like protoxin that has minimal non-target effects and has not produced strong resistance in targeted mosquito populations in >30 years of use [13]. However, since the toxin is an acellular protein complex, it is subject to inactivation and does not recycle. Additionally, in polluted or organic-rich waters its efficacy is diminished, requiring much higher amounts to kill larvae [13]. Further, applying the Bti spores is not efficacious as they do not reproduce well in most aquatic habitats. Fortunately, the elucidation of the Bti toxin at the molecular level has enabled approaches where the toxin can be delivered as an intracellular protein within a heterologous host that grows in larval habitats as a food source [14].

The advantages of using *C. reinhardtii* as the heterologous host and engineering the chloroplast genome with the toxin genes has been discussed previously [12]. We also showed that the *cry11Aa* gene and a truncated *cry4Aa* gene (*cry4Aa_700_*) could be expressed in the chloroplast at high levels and the resulting strains were toxic to mosquito larvae [12]. However, to achieve high-level expression of the cry genes, which are large proteins, we relied on an inducible system, where the transgenic mRNA was the only mRNA with a 5′ element from the *psbD* gene (because the endogenous *psbD* gene had been engineered to have a different 5′ element). This system also requires a non-wild-type host strain, Ind41_18, that has been engineered for inducible expression of the *trans*-factor for the *psbD* 5′ element, called Nac2 [15]. However, as indicated above, we need to use a robust wild-type host with more or less constitutive (chloroplast gene expression does fluctuate diurnally [16,17]) expression of the ***cry*** genes. This goal turned out to be difficult to achieve, however, and so we decided to focus on just Cry11A protein production, which is the subject of this report. Cry11A is the most potent of the Bti toxin genes [18].

## 2. Materials and Methods

### 2.1. Chlamydomonas Strains, Growth, and Chlorophyll Determination

The *C. reinhardtii* strains used for chloroplast transformation were from the *Chlamydomonas* Resource Center (University of Minnesota; https://www.chlamycollection.org). Strain CC-373 (ac-u-c 2-21) contains a deletion in the chloroplast *atpB* gene, which prevents photosynthesis; introduction of the *atpB* gene by transformation restores the gene and produces a wild-type chloroplast [19,20]. Strain CC-1690 is R. Sager’s wild-type strain, 21gr, which can utilize nitrate as the sole nitrogen source [21]. The standard medium for culture growth was Tris-acetate-phosphate (TAP) medium [21]. However, for the CC-373 transformants, minimal medium was used, which was made by adjusting the pH of TAP with HCl instead of HOAc. Cultures on plates were grown in moderate continuous light (~4000 lux) at 24 °C. For liquid cultures, flasks that were ~40% full were used with continuous shaking (~140 rpm) under moderate continuous light (~4000 lux) at 25 °C. Cell counts were inferred from total chlorophyll measurements using 4 µg total chlorophyll per 1 × 10^6^ cells as the standard reference value; total chlorophyll was measured as described previously [12]. Briefly, culture aliquots (1 mL) were centrifuged in a microfuge for 5 min, and the resulting cell pellet was extracted with 1 mL of 95% EtOH and mixing. After recentrifugation (2 min), the supernatant was removed and its absorbance at 665 and 649 nm was measured and used to calculate total chlorophyll.

### 2.2. Transgene DNA Constructs

All the expression elements used for the gene constructions are derived from plasmids containing *C. reinhardtii* chloroplast DNA sequences, and the PCR primers are listed in Appendix A. As with the previous construction of *psbD_m_*:*cry11A*:*psbA* [12], the *cry11A* expression cassettes were first assembled in plasmid pET-16b, which has a low copy number, and then transferred to the chloroplast transformation plasmids for introduction into the organelle. The assembly in pET-16b utilized restriction sites NcoI, NdeI, and BlpI in its multicloning site (MCS); however, the BamHI site in the MCS was pre-destroyed by cutting with BamHI, blunting with Klenow polymerase, and then re-ligation. This permitted the use of BamHI sites at the beginning and end of the *cry11A* transgenes, for their excision and subsequent insertion into transformation plasmids. The *cry11A* coding sequence was also described previously: it is codon-optimized for the *Chlamydomonas* chloroplast and contains a FLAG peptide at its C-terminus [12]. It was synthesized commercially (Integrative DNA Technologies) and provided in a plasmid flanked by NdeI and XbaI sites. After cutting with XbaI, blunting, then cutting with NdeI, it was ligated into the modified pET-16b (minus the BamHI site), which had been cut with XhoI, blunted, and cut with NdeI; the new plasmid was *cry11A*-pET-16b. In general, the 3′ element was added first to the *cry11A* coding region and then the 5′ element(s) (either promoter-UTR or promoter + 5′ UTR) were added second. The oligo primers for the PCR amplifications are listed in Appendix A; the reactions also contained recombinant plasmids as templates and the enzyme was the Phusion DNA polymerase (New England Biolabs, Ipswich, MA, USA). The products were restricted and ligated to the *cry11A* transgene. The final transgene clones were sequenced to verify the accuracy of the various fusions and ligations.

The 3′ UTR-containing element that was added to *cry11A* was from the *r**bcL* gene [22]. The forward PCR primer (882 in Appendix A) contained a BlpI site at its 5′ end, while the reverse primer (883) contained BamHI and BlpI sites at its 5′ end. The PCR product was cut with BlpI, purified on a PureLink spin column (Invitrogen, Waltham, MA, USA), and ligated to the *cry11A*-pET-16b plasmid cut with BlpI and treated with Antarctic phosphatase; the BlpI restriction site is directional so that the DNA inserts only in the forward direction.

The *rps4* gene is co-transcribed with and immediately downstream of the tRNA gene, *trnG* [23,24]. Thus, to use the native promoter and 5′ UTR for *rps4*, we amplified the same region as in Fargo et al. [25], which spans from 229 nt upstream of the *trnG* gene through the *rps4* 5′ UTR. The plasmid template was P-657 [25], the forward primer (997) contained NcoI and BamHI sites at its 5′ end, and the reverse primer (998) contained an NdeI site at its 5′ end, which inserted 3 nt (CAT) just before the ATG start codon on the sense strand. The PCR product was digested with NcoI and NdeI, and ligated to the *cry11A:rbcL* gene plasmid cut with the same enzymes, and treated with Antarctic phosphatase. 

In order to add a native 16S *rrn* promoter (16S_N_), nt −91 to −1 [26], and the 5′ UTR of the mature *rps4* mRNA to *cry11A*:*rbcL*, the 16S promoter was amplified with forward primer 1007 and a long reverse primer (1022) that contained the 3′ end of the promoter plus the entire mature *rps4* 5′ UTR, which is only 42 nt [24]. The forward primer also contained, at its 5′ end, NcoI and BamHI sites, and the large reverse primer contained at its 5′ end an NdeI site, which inserted CAT just before the ATG start codon on the sense strand. The PCR product was cut with NcoI and NdeI and ligated to the *Cry11A:rbcL* gene plasmid, which had been cut with NcoI and NdeI and treated with Antarctic phosphatase.

To add the *psbD* promoter and *rps4* 5′ UTR to the *cry11A*:*rbcL* gene, the forward primer (847) was located >100 nt upstream of the *psbD* promoter, and the long reverse primer (1021) contained the 3′ end of the *psbD* promoter plus the entire mature *rps4* 5′ UTR. As in the preceding case, the forward primer also contained at its 5′ end NcoI and BamHI sites, and the large reverse primer contained at its 5′ end an NdeI site, produced by insertion of CAT just before the ATG on the sense strand. Ligation of the NcoI/NdeI-cut PCR product to the *cry11A:rbcL* gene was also as described above.

### 2.3. Construction of the Plasmids for Biolistic Transformation of the Chloroplast

Derivatives of plasmid p322: plasmid p322 was constructed by Newman et al. [27] and contains the region of the *Chlamydomonas reinhardtii* chloroplast genome from the EcoR1 site in *psbA* intron 4 to the XhoI site in the *Cr.LSU* intron of the 23S rRNA gene. We chose the unique BamH1 site in the intergenic region between *psbA* and 5S as the site for insertion of the c*ry11A* transgene, but first a second BamHI site in the MCS of p322 was removed as previously described to give p322.1 [12]. Next, the recyclable selectable marker 483-*atpA-aadA-rbcL*-483 [28] was inserted into p322 as previously described [29]; this allows selection of transformants by spectinomycin resistance. For insertion of the *cry11A* transgene into the BamHI site of p322.1, they were excised from the pET-16b constructs by cutting with BamHI, purified by GTG agarose gel electrophoresis, and then ligated into BamHI-cut, Antarctic phosphatase-treated p322.1 (containing the 483-*atpA-aadA*-*rbcL*-483 marker) with T4 DNA ligase. This allows the insert to be incorporated in either forward or reverse orientation.

Derivatives of plasmid p655: plasmid p655 were designed by Fargo et al. [25] for biolistic transformation of *C. reinhardtii* strain ac-u-c 2-21 (CC-373). This strain has a single deletion that includes a large portion of the *atpB* gene and extends into a significant portion of the adjacent large inverted repeat (IR) [20]. Plasmid p655 contains the 5.25 kb BamHI-EcoRI chloroplast DNA fragment that encompasses the deletion and includes at least 1 kb of homology on either side, which enables efficient recombination with the genome. Recombination repairs the *atpB* gene and transformants can be selected by plating on minimal media. p655 also includes a cloning site, flanked by XbaI and XhoI restriction sites, located about 495 nt upstream of the 3′ end of the large IR. The original p655 also contained an *aadA*-based transgene expression cassette in the cloning site, but we removed this entire cassette by cutting with XhoI and XbaI, followed by followed by blunting with Klenow fragment, dephosphorylation with Antarctic phosphatase, and purification by GTG agarose electrophoresis. A given *cry11A* transgene was then excised from its pET-16b home by cutting with BamHI, followed by blunting with Klenow, purification, and ligation to the pre-treated p655 fragment. The transgene insert can be incorporated in either the forward or reverse direction.

### 2.4. Biolistic Transformation, Selection and Identification of Homoplasmic Transformants

Biolistic transformation [19] was performed using S550d gold particles (Sea Shell Technologies, La Jolla, CA, USA) essentially as described by the manufacturer using a protocol optimized for *Chlamydomonas*. For each biolistic sample, 1 mg of gold particles and 2 µg of plasmid DNA were used. The p322-derived plasmids were shot into strain CC-1690, and the p655-derived plasmids were shot into strain CC-373. The Bio-Rad PDS-1000 biolistic apparatus was used with 1100 psi rupture discs. The agar plates used for shooting contained TAP medium plus 100 µg/mL ampicillin (TAP + Amp) to minimize bacterial contamination. For each plate, 4 × 10^7^ cells in 0.8 mL of TAP/0.125% agar were pipetted onto the middle of the plate in a circle of ~2.8 cm diameter and allowed to dry to a flat sheen. After shooting, the plates were kept in very dim light for 20–24 h and then transferred to selection plates, which were TAP-minimal for CC-373 transformants and TAP + 100 µg/mL spectinomycin (TAP + Spec) for CC-1690. About 70–80% of the cells from the shooting plates were transferred to the new plates, which were incubated at 24 °C in bright light until untransformed cells had died and colonies of transformants had appeared, usually in about 10–15 d.

Transformant colonies were transferred to fresh plates, TAP for CC-373 and TAP + Spec for CC-1690, spread into about 0.5 cm squares and incubated in bright light until vigorous growth had occurred, at which time part of each square was used for DNA extraction as previously described [12,30]. The DNA was used to test for homoplasmicity by PCR with primers that bind on either side of the transgenic DNA insertion site. For the CC-373 transformants, the primers were 1008 and 1009, which give a PCR product of 633 bp if there is no transgene (and about 3.5 kb if the insert is present). For the CC-1690 transformants, primers 863 and 975 were used, which give a PCR product of 124 bp if there is no insert and about 5.3 kb if the insert is present. Obtaining homoplasmic transformants typically required multiple rounds of colony selection and screening, as follows: colonies with the greatest amount of large PCR product were resuspended (at low cell concentration) in sterile TAP medium and 10 µL of this suspension was streaked with a sterile wire loop over an entire new plate, which was then incubated in bright light until new colonies appeared. These were then screened by PCR to identify those that had lost the small PCR product (i.e., the DNA insertion site) entirely, indicative of a complete replacement of the recipient genome with the transformed genome. Usually 2–3 rounds of this subcloning were sufficient to give homoplasmicity for the transgene insert. 

### 2.5. Protein Extraction and Western Blotting 

Transformants were grown in TAP medium (50 mL) to early stationary phase (5–7 × 10^6^ cells/mL) and then for another 24 h before harvesting (cell density was re-checked just before harvesting but had increased <10% from the day before). Total protein was calculated from total chlorophyll measurements using the approximate conversion, 1 µg chlorophyll = 10 µg total protein, which we had previously determined for these strains using the Bradford assay for protein determination [12,29]. Cells were then pelleted at room temperature for 5 min at 2600× *g* and resuspended in 0.6 mL of 100 mM Tris-HCl pH 8.5, 100 mM DTT, 7 mM benzamidine, 5 mM EDTA (Resuspension Buffer or RB), and then transferred to 1.7 mL microfuge tubes. The vol was adjusted to 0.8 mL with RB and then 0.2 mL of 10% SDS was added, to give a final concentration of 2% SDS and a final vol of 1 mL. The samples were put on a rocker for 10 m at room temperature to solubilize the cells, and the lysates frozen at −80 °C. The expression in *E. coli* and purification of rCry11A, which has an N-terminal his-tag as well as the C-terminal FLAG tag, was as described previously [12].

For SDS-PAGE, the lysates were thawed, thoroughly mixed at room temperature, and 5 µL aliquots withdrawn to new microfuge tubes. Then, 2% SDS was added to these aliquots to give a final total protein concentration of 1.5 µg/µL, followed by addition of 0.5 vol of 3× SDS loading buffer (6% SDS, 50 mM Tris-HCl pH 6.8, freshly added 0.3 M DTT, 0.03% bromophenol blue, 30% glycerol), to give a final total protein concentration of 1 µg/µL. The samples were heated 5 m at 100 °C and then typically 10 µL (10 µg protein) were loaded per lane on a 10% polyacrylamide gel along with a prestained protein ladder (Precision Plus, Bio-Rad, Hercules, CA, USA) as size markers (usually, a second identical gel was loaded and run simultaneously in a Hoeffer MiniGel apparatus providing one for protein staining and one for blotting.) After electrophoresis, gels were soaked in cold transfer buffer (25 mM Tris base, 192 mM glycine, 5% methanol) for 15 m and then electrotransferred to a PVDF membrane (Hybond-P, GE HealthCare, Chicago, IL, USA) using a Genie blotter (Idea Scientific, Minneapolis, MN, USA) at 12 V for 2 h at 4 °C [31]. The membrane was then blocked overnight at 4 °C with 5% nonfat dried milk in TBS-T (20 mM Tris-HCl pH 7.6, 150 mM NaCl, 0.05% Tween-20). After washing twice for 5 min with 0.5% nonfat milk (in TBS-T), the membrane was incubated for 2 h (at room temperature) with an anti-FLAG monoclonal antibody (M2) coupled to alkaline phosphatase (Sigma-Aldrich A9469, St. Louis, MO, USA) that had been diluted 1:5000 in TBS-T plus 0.5% nonfat milk. Then, the membrane was rinsed twice (2 min each) with 30 mL of 0.5% nonfat milk in TBS-T, and then four times (10 min each) with 50 mL of TBS-T. Finally, the membrane was rinsed with purified water for 5 min, overlaid with 2 mL of CDP-Star AP substrate (EMD Millipore Corp, Billerica MA, USA), and incubated for 3 min. The substrate was decanted, the edges of the membrane blotted with a Kimwipe (Kimberley-Clark, Irving, TX, USA) and then it was put in a sheet protector bifold to prevent drying. After 40 m, the membrane was exposed to X-ray film for periods ranging from 1 to 30 s.

### 2.6. Reverse-Transcription PCR (RT-PCR)

First, 50 mL cultures of transformants were grown to a cell density of about 3 × 10^6^ cells/mL. Then, 10 mL of each were pelleted and total nucleic acids (TNA) purified as previously described [30]; the TNA concentration was measured by absorption at 260 nm. For the RT-PCR, 10 µg of each TNA preparation was treated with Turbo DNase (Ambion, Austin, TX, USA) in a total vol of 50 µL (as described by the manufacturer) to destroy the DNA. Then, 4 µL of the DNase-treated samples were incubated at 50 °C for 1 h with reverse transcriptase (Superscript III, Invitrogen) exactly as recommended by the manufacturer. The total reaction vol was 20 µL and the gene-specific reverse primer was 800 for *cry11A* mRNA. For the PCR step, 0.5 µL aliquots from the RT reactions were amplified with *Taq* DNA polymerase (NEB) in a total vol of 25 µL using primers 800 and 1002 [12]. The number of PCR cycles was determined by trial and error and ranged from 23 to 27 [12], as we wanted the product yield to still be increasing. Minus–RT reactions on all the preparations confirmed the dependence of the PCR products on the reverse transcription step. After PCR, the products were analyzed on 1.5% agarose gels and the DNA-ethidium fluorescence was photographed with a digital camera (Bio-Rad) and the image was inverted. 

### 2.7. Mosquito Bioassay

For the live cell bioassay with mosquito larvae the indicated transformants and control (a CC-373 transformant copy-corrected to contain no Cry gene but still having a repaired *atpB* gene) were grown in TAP medium to essentially stationary phase (about 5–7 × 10^6^ cells/mL) and then another 24 h before harvesting. The cells were pelleted by centrifugation, washed once with 40 mL of distilled water, resuspended in distilled water to 4 × 10^6^ cells/mL, and then diluted with water as necessary to give the final desired cell numbers. For each assay, 20 mL, containing twice the desired final concentration of algae, were added to 20 mL of distilled water containing 12 3rd instar *Aedes aegyptii* (Galveston, TX, USA) larvae. These larvae were raised on a mosquito diet (Carolina Biological, Burlington, NC, USA) and then starved for 48 h prior to the assay. Plastic cups were used for the assay and they were covered with clear plastic wrap secured with a rubber band. The samples were incubated at 27 °C under a 12 h:12 h light–dark cycle and were monitored daily for 5 d.

## 3. Results

### 3.1. Poor Expression of the psbD_m_:cry11A:psbA Transgene in a Wild-Type Background

The goal of this research was to achieve a high level of rCry11A production from the chloroplast genome in an otherwise wild-type strain of *C. reinhardtii*. We previously obtained high-level expression of rCry11A with the inducible system, in which the *cry11A* gene is expressed from a *psbD_m_:cry11A:psbA* transgene integrated into the IR (inverted repeat) region of strain Ind41_18 [12]. To test this transgene’s expression in a wild-type background, the same *psbD_m_:cry11A:psbA* construct was integrated into the IR region of two wild-type strains, 2137 and 21gr, at a BamHI site between the *psbA* and 5S rRNA genes (Figure 1A); the adjacent *atpA*:*aadA*:*rbcL* gene provided the selection mechanism by conferring spectinomycin resistance on the transformants. The results were similar in both host strains, and those with the 21gr strain are shown in Figure 1. Figure 1A also shows how we use PCR to determine when the chloroplast transformants become homoplasmic, i.e., we look for a complete loss of the target DNA site as all copies of cpDNA become replaced with the copies containing the inserted transgene. This was the case for all of the *psbD_m_:cry11A:psbA* transformants analyzed in Figure 1A (lanes 2–5); the negative control strain lacked the transgene insertion (lane 6). Finally, as indicated in the figure, the *psbD_m_:cry11A:psbA* transgene was introduced into the genome in both orientations.

Homoplasmic transformants for each orientation of the *psbD_m_:cry11A:psbA* transgene were grown in liquid TAP media under continuous light, our standard conditions, and examined for rCry11A (recombinant Cry11A) production by Western blotting. The FLAG tag at the C-terminus of rCry11A allowed us to employ a commercial anti-FLAG, enzyme-conjugated monoclonal antibody to detect the protein, which we did after separating total protein on a SDS gel and transferring it to a membrane (Western blot). Figure 1B shows that levels of rCry11A were very low in both transformants (lanes 2 and 3), being barely detected even though 30 µg of total protein were loaded compared to only 10 µg for the Ind41_18 strain (carrying *psbD_m_:cry11A:psbA*) under inducing conditions (lane 1). With the latter strain under the inducing condition, rCry11A is 0.35–0.4% of total protein [12] and this is our benchmark strain. These results indicate that the *cry11A* coding region is recalcitrant to expression in a wild-type setting, because the *cyt1A* coding region, also from the Bti toxin [13], was highly expressed in wild-type using these same 5′ and 3′ elements (i.e., *psbD_m_* and *psbA*, respectively) [29].

### 3.2. Attempts with 16S-atpA, atpA, and rbcL 5′ Elements on the cry11A Transgene in Wild-Type

One of the very few ways that high-level expression of a transgene in a wild-type chloroplast has been achieved is by using the strong 16S rRNA promoter and the 5′ element from the *atpA* gene to drive expression [4,32,33]. Hence, we tried using the same 16S promoter region and 5′ element from *atpA* used previously [4,32,33] for expressing *cry11A*. We also used a different transformation system that was more convenient and avoided the need for a selectable marker, which, with its expression elements, would have added yet another copy of a homologous 5′ element. Briefly, we used the CC-373 strain as host, which has a deletion that includes a large portion of the *atpB* gene (see Figure 2A below). When plasmid P-655 derivatives are transformed into its chloroplast, the deletion is repaired, restoring the wild-type genome and photosynthesis; transformants are selected by plating on minimal media. Replacing the *psbD_m_* 5′ element (promoter and 5′ UTR) on *psbD_m_:cry11A:psbA* with the 16S *rrn* promoter and 5′ UTR from *atpA*—creating *16S:atpA:cry11A:psbA*—did not improve expression of the rCry11A protein. Similarly, replacing *psbDm* with both the promoter and 5′ UTR from *atpA* (*atpA:cry11A:psbA*) and *rbcL* (*rbcL:cry11A:rbcL*) also did not improve expression of the rCry11A protein (data not shown).

### 3.3. Using the rps4 5′ UTR with Its Native Promoter to Drive cry11A Expression in Wild-Type

As discussed above, nearly all bioengineering studies to date have used elements from highly expressed photosynthesis genes for transgene expression in the *C. reinhardtii* chloroplast. However, photosynthesis genes appear to be under tight nuclear and auto-regulatory control. Since it is not clear if this model applies to chloroplast ribosomal protein genes; we considered the possibility that they are not as tightly controlled with specific trans-factors. Moreover, the Boynton–Gillham lab showed that *rps4* and *rps7* 5′ elements drive expression in *E. coli* [25], which does not have specific factors for chloroplast mRNAs, suggesting their mRNAs may have inherent affinity for ribosomes. We were particularly attracted to *rps4* because the predicted 5′ UTR of mature *rps4* mRNA is quite short (42 nt), lacks a stable secondary structure, and the very 5′ end corresponds to a small RNA (sRNA) of 29 nt [24], suggestive of a binding site for an end-protecting protein.

*Rps4* is downstream of, and co-transcribed with, the transfer RNA gene, *trnG* [24]. To determine if it could drive *cry11A* expression, we initially employed its whole 5′ element, which included the native promoter upstream of *trnG* and the *trnG-rps4* intergenic region, similar to Fargo et al. [25]. The *trnG-rps4:cry11A:rbcL* transgene was integrated into the chloroplast IR in strain CC-373 as described above. The PCR analysis shows that the transformant analyzed in Figure 2 was very close to homoplasmic but may have had a small amount of the recipient genome left (Figure 2A). Nonetheless, rCry11A was readily detected on the Western blot (lane 1), though its level is substantially less than our benchmark strain (*psbD_m_:cry11A:psbA* transgene in the inducible Ind41_18 host) (Figure 2B, lanes 2 and 3). Additionally, since some of our purified rCry11A standard was included on the gel (Figure 2B, lanes 4 and 5), this result confirms that the rCry11A level in the induced Ind41_18 strain (lane 2) is ~80 ng per 20 µg, or 0.4% total protein, as described previously [12]. The production of Cry11A in the corrected wild-type strain with the *trnG-rps4:cry11A:rbcL* transgene appeared promising, since we had not yet optimized the elements upstream of the *rps4* 5′ UTR.

### 3.4. Using the 5′ UTR from Mature rps4 mRNA with Different Promoters to Drive cry11A Expression in a Wild-Type Background

We decided to try two things to improve rCry11A production: (1) to use the 5′ UTR from mature *rps4* mRNA without any upstream transcript sequences that might need processing (Figure 3), and (2) to couple it to a strong promoter. In this regard we decided to try the promoters for *psbD* and for 16S *rrn*. It should be noted that the 16S promoter used previously [4,32,33] actually contained 70 nt of downstream sequence, which would have normally been part of the pre-16S rRNA [26]. Since we were not certain of the possible effects that this RNA sequence might have on processing and/or translation, it was not included; hence, we made a clean fusion of the native 16S promoter (16S_N_, nt −91 to −1) (Figure 3) to the **mature** *rps4* 5′ UTR. The construction process did create an additional triplet right before the start codon of *cry11A*, CAU (Figure 3), but this preserved a U nt at −1, which the native rps4 UTR also has and could be involved in extended pairing to the initiator tRNA^fmet^ [34]. We also created inadvertently a 16S promoter with a −1 deletion (referred to as 16S_m_ in Figure 3), and this was tested as well, after fusing it to the mature *rps4* 5′ UTR. The *psbD* promoter shown in Figure 3 was also fused to the mature *rps4* 5′ UTR. All three elements were fused to the *cry11A:rbcL* construct, and transformed into the CC-373 chloroplast, in both orientations (forward, F and reverse, R).

Homoplasmic transformants were obtained for each construct, and the level of rCry11A was examined by Western blot analysis of equal amounts of total protein (10 µg) for each type (Figure 4: lanes 1–2, 4–7). Protein from the previously analyzed (see Figure 2) *trnG-rps4:cry11A:rbcL* transformant (Figure 4, lane 3) and from our benchmark strain, *p**sbD_m_:cry11A:psbA*/Ind41_18 grown under inducing conditions (Figure 4, lane 8), were also included for comparison. The results show that orientation only mattered for the *16S_M_:rps4:cry11A:rbcL* transgene, whose expression was lower than either the *psbD:rps4:cry11A:rbcL* or *16S_N_:rps4:cry11A:rbcL* transgenes; the latter of which gave the highest level of rCry11A. In fact, the rCry11A production from the *16S_N_:rps4:cry11A:rbcL* transgene in this wild-type background exceeded that of our benchmark strain (*p**sbD_m_:cry11A:psbA*/Ind41_18 induced) by about 50%, reaching ~0.6% of total protein. It is also remarkable that, by using the mature 5′ UTR from *rps4* as the 5′ UTR, the expression level of the rCry11A protein became largely promoter-dependent, suggesting that the post-transcriptional bottleneck had been removed.

### 3.5. RT-PCR Analysis of the New cry11A Transgenes Expressed in Wild-Type Chloroplast

RT-PCR was performed to get a semi-quantitative evaluation of RNA levels for these transgenes in wild-type transformants. The results in Figure 5 show that relative transcript levels correlate with the rCry11A protein levels in that *16S_N_:rps4:cry11A:rbcL* RNA levels exceed *psbD:rps4:cry11A:rbcL* RNA levels which are greater than *16S_M_:rps4:cry11A:rbcL* RNA levels. Somewhat surprisingly, the *trnG-rps4:cry11A:rbcL* transcript level, which was analyzed in parallel (Figure 5, lane 7) is similar to the *psbD:rps4:cry11A:rbcL(R)* level, though not quite as strong as the *psbD:rps4:cry11A:rbcL (F)* level. In contrast, the expression of rCry11A protein in the *trnG-rps4:cry11A:rbcL* transformant was several fold lower than the *psbD:rps4:cry11A:rbcL* transformants, suggesting that RNA from the *trnG-rps4:cry11A:rbcL* transgene may be poorly translated. These results are also congruent with the protein data for the mutant 16S promoter (Figure 4) transgene, *16S_N_:rps4:cry11A:rbcL*; thus, confirming the importance of the −1 position in the 16S *rrn* promoter. 

### 3.6. Toxicity to Aedes aegypti Larvae of the 16S_N_:rps4:cry11A:rbcL Strains

In order to assess the toxicity of the best rCry11A-producing strains, we used a bioassay with live algae and *Aedes aegypti* larvae (3rd instar) incubated in dH_2_0. With these conditions, the algal cells do not grow or divide, so the cell numbers per mL are the cell concentrations at the beginning of the incubation period (i.e., initial cell concentrations); they decline as the larvae consume them. Table 1 summarizes two of the bioassay experiments, which were carried out at decreasing initial cell numbers (preliminary assays indicated that all the larvae were killed at high initial cell numbers (>1 × 10^6^ cells/mL)). The data show that even with initial cell numbers as low as 2 × 10^4^ cells/mL, most of the larvae were killed within 72 h, especially for the *16S_N_:rps4:cry11A:rbcL* (F) strain. To provide a visual display of the algal cells at these cell concentrations, flasks of *C. reinhardtii* at 2 × 10^6^, 2 × 10^5^, and 2 × 10^4^ cells/mL are shown in Figure 6. The flask with the lowest cell concentration (2 × 10^4^ cells/mL) barely shows any green at all, and in this respect almost resembles tap water. This reinforces the conclusion that these rCry11A-producing strains are highly toxic to *Aedes aegypti* larvae. 

### 3.7. Stability of the 16S_N_:rps4:cry11A:rbcL Strains: Visualization of rCry11A by Coomassie Staining

Stability of transgenes is an important issue for the potential employment of such strains for mosquito control. DNA analysis by PCR over a period of ~2 yrs from their creation did not show evidence of loss or rearrangement of the *cry11A* transgenes, including in the *16S_N_:rps4:cry11A:rbcL* (F) and *16S_N_:rps4:cry11A:rbcL* (R) strains (not shown). Moreover, analysis of rCry11A protein levels in these strains showed little to no evidence of declining over time (Figure 7); the two *16S_N_:rps4:cry11A:rbcL* (F) preparations probed in Figure 7 were prepared 15 mos apart, and the *16S_N_:rps4:cry11A:rbcL* (R) preparation was made >18 mos after the initial analysis of these strains (see Figure 3).

Figure 7 also shows that rCry11A can be detected by Coomassie staining if the SDS-PAGE is performed on an 8% polyacrylamide gel run for an extended period (the small proteins run off). All three of the transgenic preparations show an enhanced Coomassie-stained band of the right size (73 kDa) which co-migrated exactly with the anti-Flag antibody signal (lanes 2–4 and 2′–4′); whereas the control strain lacking the *cry11A* transgene did not (lanes 1 and 1′). Being able to detect transgene products by Coomassie staining of total algal proteins on mini-gels is unusual, especially for a relatively large protein (73 kDa) such as rCry11A. It also makes for a more convenient test of rCry11A production than Western blots.

## 4. Discussion

We previously reported high level expression of this same codon-adapted *cry11A* gene in the chloroplast of *C. reinhardtii* using the inducible *psbD*/Nac2 system [15]. However, when the same *psbD:cry11A:psbA* transgene was introduced into the chloroplast of wild-type strains, expression of the rCry11A protein was very low. Changing the 5′ elements to *atpA*, *rbcL*, or 16S-*atpA* did not improve the protein expression, even though the 16S-*atpA* combination had worked for others [4,32,33]. We decided to try a new tack and get away from photosynthesis genes, and so we tested the *rps4* 5′ element, specifically the 5′ UTR from mature *rps4* mRNA, and did not include any upstream sequences that might need to be processed. This worked very well, producing high levels of rCry11A protein when coupled to either the *psbD* or 16S *rrn* promoter. The 16S promoter was a little more robust, however, and the rCry11A protein levels obtained with *16S_N_:rps4:cry11A:rbcL* in a wild-type background are ~50% greater than we obtained with the inducible system, or ~0.6% versus 0.4% of total protein [12]. It should also be said that rCry11A is a relatively large protein, at 73 kDa, and most reports of robust expression of genes in the chloroplast are relatively small proteins (e.g., [5,6,7]). In that regard, we did obtain high-level expression of the 27 kDa Cyt1A toxin in wild-type using the same expression elements that we used in the inducible system, i.e., *psbD:cyt1A:psbA*. Perhaps larger coding regions present more potential problems for the transgene by having more potential for misfolding, i.e., folding into alternative structures that impede the function of the 5′ elements in promoting mRNA stability and translation. 

The bioassay indicated very high toxicity of these strains to *Aedes aegypti* larvae, killing most larvae even at low cell numbers (2 × 10^4^ cells/mL). It should be noted that by performing the bioassays in dH_2_0, the algae do not grow during the assay and cell number actually declines during the incubation period as the algae are consumed. However, this condition enables us to compare to some other published studies, such as those with engineered cyanobacteria [14,35,36]. Indeed, these *C. reinhardtii* strains appear to be as toxic as the most toxic cyanobacteria expressing Bti proteins [35,36].

The bottleneck for transgene expression, especially in a wild-type background, is usually taken to be post-transcriptional. These data support that conclusion, at least for the photosynthesis genes which have been the primary source of expression elements, since employing the mature 5′ UTR of *rps4* on the *cry11A* transgene made rCry11A production largely promoter-dependent. The *rps4* 5′ UTR element was attractive because: (1) it had already been shown that *rps4* is translated in *E. coli*, which should not have specific translation factors for this chloroplast gene; (2) the first 29 nt of the mature UTR corresponds to a small RNA [24], which could indicate that it is a protein binding site (and possibly stabilizing for the 5′ end of the mRNA); and (3) the mature UTR is relatively small overall, 42 nt, and lacking in secondary structure. How large the role that any of these factors, or the fact that *rps4* is a ribosomal protein (and not a photosynthetic protein), played in alleviating the block in *cry11A* expression is unknown. However, the fact that the putative protein binding site is in the first 29 nt suggests that a protein(s) could bind that site before the remainder of the mRNA is synthesized; thus, avoiding problems that could come from unfavorable interactions of the 5′ UTR with downstream sequences.

A surprising result was that a single nucleotide deletion at −1 in the 16S promoter significantly compromised expression. This deletion, which was inadvertently introduced during one of the 16S promoter amplifications, effectively changed the −1 nt from C to G and the +1 nt from G to T (Figure 3). The roles and/or importance of the nts at the initiation site have not been evaluated for chloroplast promoters to our knowledge, but the identity of the +1 nt has been shown to be important for transcriptional regulation in *Bacillus subtilis* [37]. Interestingly, this seems to involve the relative concentrations of free ribonucleotide triphophates, ATP or GTP, that are used to start the RNA. Since previous researchers used a version of the 16S promoter that included 70 nt of downstream sequence [4,32,33], the −1 and +1 positions would have been the same as the native 16S sequence. However, we wanted to avoid having an RNA sequence at the beginning of the transcript that might have unintended effects, so we cut the promoter off at the −1 nt.

Finally, the *16S_N_:rps4:cry11A:rbcL* strains also appear to be very stable. We have not seen evidence of transgene loss or DNA rearrangements, or for that matter any evidence of a loss of expression of rCry11A protein for at least 2 years.

## 5. Conclusions and Future Direction

This work represents a milestone in our quest to develop strains of *C. reinhardtii* that could be used for pest control, not only for controlling disease-transmitting mosquitoes and black flies [13] but possibly also for control of gastrointestinal nematodes [38]. The toxicity of these strains against *Aedes aegypti* larvae, which is a valid indicator, is high enough such that only low cell numbers should be needed for control of the mosquitoes (*Aedes* and *Culex*) that transmit viruses such as Zika and West Nile. This is important because dominance in mosquito larval habitats over other microalgae would probably not be sustainable with *C. reinhardtii*. We are cognizant of the fact that mosquitoes have developed resistance to Cry11Aa in the laboratory setting, but that the addition of Cyt1Aa overcomes that resistance [13]. To that end, we have already developed a construct that expresses rCyt1Aa well in the chloroplast of wild-type strains [29], and we have recently made some strains that express both rCry11A and rCyt1A proteins. These latter strains are not yet completely stable, and work on them is ongoing. 

## Figures and Tables

**Figure 1 microorganisms-10-01087-f001:**
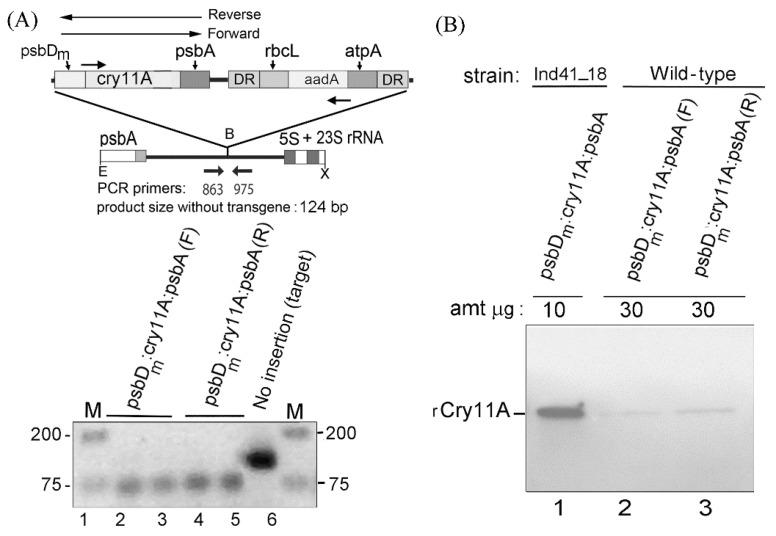
Transformation and expression of the *psbD_m_:cry11A:psbA* transgene in a wild-type *C. reinhardtii* strain (CC-1690). (**A**) Transforming DNA and test of homoplasmicity by PCR. (*Above*) The diagram is of the introduced DNA (top, expanded), and the site of integration in the chloroplast genome at the BamHI site. The selectable marker gene is *atpA:aadA:rbcL,* and it is flanked by a direct repeat (DR) that allows for loss of the marker gene by recombination, though that was not sought in this case. The location of the PCR primers used to check for homoplasmicity, and the predicted product size (124 bp) if there are no transgenes, is shown. E, B, X are restriction sites EcoRI, BamHI, and XbaI, respectively. (*Below*) PCR analysis of four transformants (lanes 2–5) and the recipient strain (lane 6); two of the transformants contained the *psbD_m_:cry11A:psbA* transgene in the forward orientation (lanes 2–3) and two had it in the reverse orientation (lanes 4–5). Lanes 1 and 7 contained DNA markers (M). The photograph is of the ethidium-DNA fluorescence gel, and the image was inverted. (**B**) Western blot analysis of *psbD_m_:cry11A:psbA* transformants, one in each orientation. In total, 30 µg of total protein was loaded for each of the *psbD_m_:cry11A:psbA* transformants (lanes 2–3), but only 10 µg of total protein was loaded for the Ind41_18/*psbD_m_:cry11A:psbA* strain grown under inducing conditions (lane 1). The blot was probed with the enzyme-conjugated anti-Flag tag monoclonal antibody.

**Figure 2 microorganisms-10-01087-f002:**
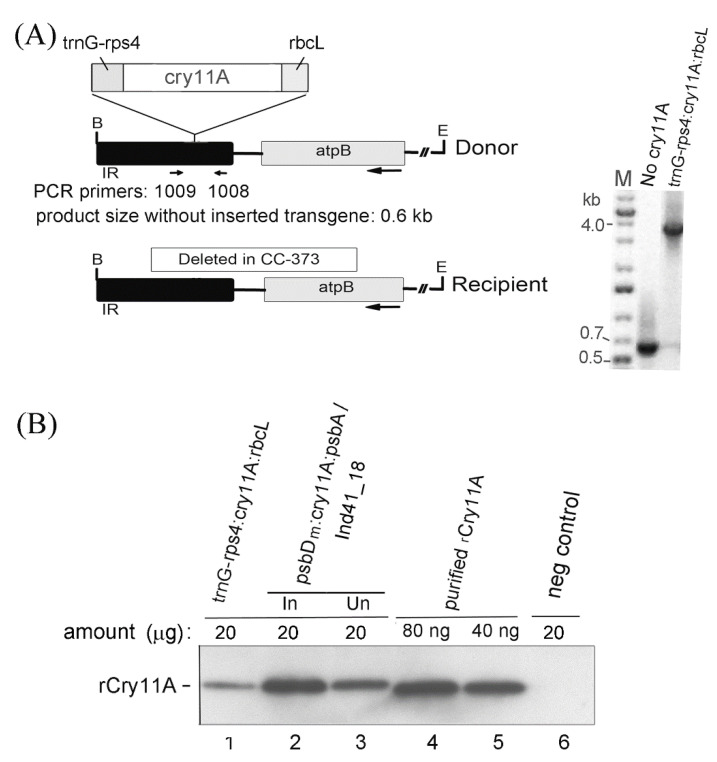
Transformation and expression of the *trnG-rps4:cry11A:rbcL* transgene in strain CC-373, which is complemented to wild-type by transformation. (**A**) (Left) Diagram of the transforming DNA (Donor) and the corresponding region of chloroplast DNA that is the homologous target in CC-373 (Recipient). The location of the PCR primers used to assess homoplasmicity is indicated. (Right) Gel of the PCR products from a *trnG-rps4:cry11A:rbcL* transformant and a transformant that did not integrate the transgene (No cry11A); lane M contained DNA size markers. (**B**) Western blot analysis of the *trnG-rps4:cry11A:rbcL* transformant from (**A**). Other lanes contained total protein from the inducible Ind41_18/*psbD_m_:cry11A:psbA* strain grown under induced (In) and uninduced (Un) conditions (lanes 2 and 3, respectively), purified rCry11A produced in *E. coli* (lanes 4 and 5), and a transformant lacking the transgene (neg control, lane 6). For the total protein samples (lanes 1–3 and 6), 20 µg of protein were loaded per lane as indicated.

**Figure 3 microorganisms-10-01087-f003:**
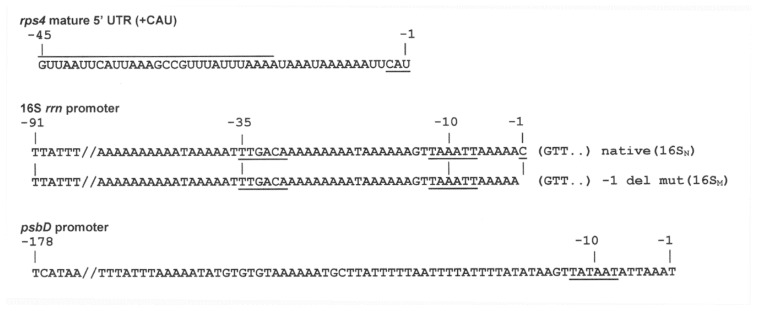
Nucleotide sequences of the mature *rps4* 5′ UTR, and the 16S *rrn* and *psbD* promoter sequences that were used in the transgene constructs. *rps4 mature 5′ UTR (+CAU)*: the 5′ UTR of mature *rps4* mRNA, plus the CAU sequence (underlined) added to the 3′ end during gene construction, is shown. The overlined segment corresponds to a sRNA [24]. The numbering is based on the A nt of the start codon for *cry11A* being +1, and so the CAU is −3 to −1. 16S *rrn* promoter: the top DNA sequence is the native 16S promoter (16S_N_), and the lower is the −1 deletion mutant (16S_M_). The −10 and −35 elements are underlined, and the first 3 nt of the *rps4* 5′ UTR sequence (GTT..), which would be numbered +1–+3 for the native 16S construction are in parentheses. *psbD* promoter: the *psbD* promoter DNA sequence was, like the 16S promoter sequences, fused directly to the mature *rps4* 5′ UTR sequence. The −10 element is underlined.

**Figure 4 microorganisms-10-01087-f004:**
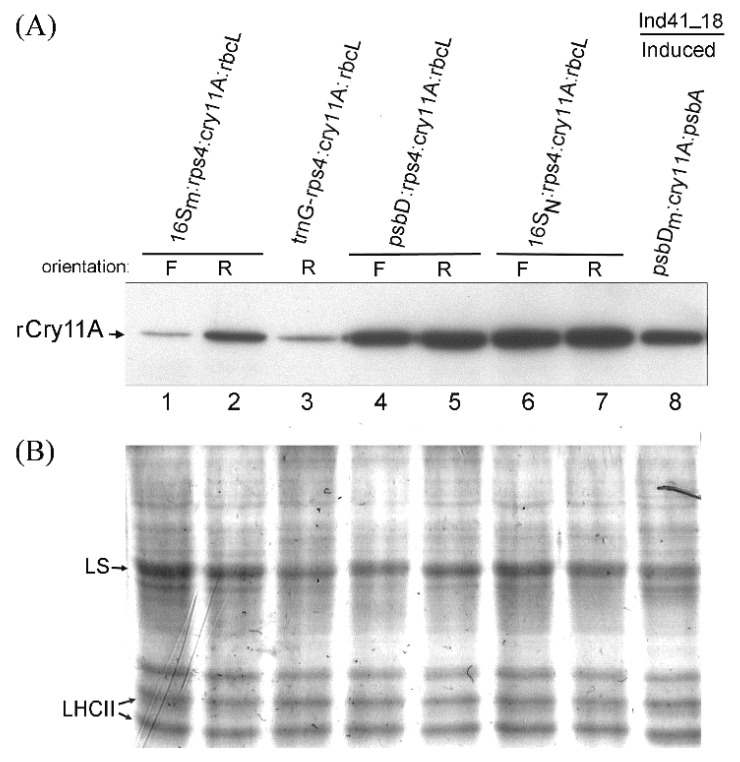
Western blot analysis of expression of the *rps4:cry11A:rbcL*-based transgenes with different promoters. The indicated transgene constructs were introduced into CC-373 in both orientations, except *trnG-rps4:cry11A:rbcL*, which was only obtained in one orientation (lane 3). Total protein (10 µg per lane) from each CC-373 transformant and from the induced Ind41_18/*psbD_m_:cry11A:psbA* strain was electrophoresed on two identical mini-gels of 10% polyacrylamide, one of which was blotted (**A**) and the other stained with Coomassie blue (**B**). The blot (**A**) was probed with the enzyme-conjugated anti-Flag antibody to detect the 73-kDa rCry11A protein. Some major chloroplast proteins, LS (large subunit of ribulose-bisphophate carboxylase) and LHCII (light-harvesting complex II) are indicated next to the Coomassie-stained gel (**B**).

**Figure 5 microorganisms-10-01087-f005:**
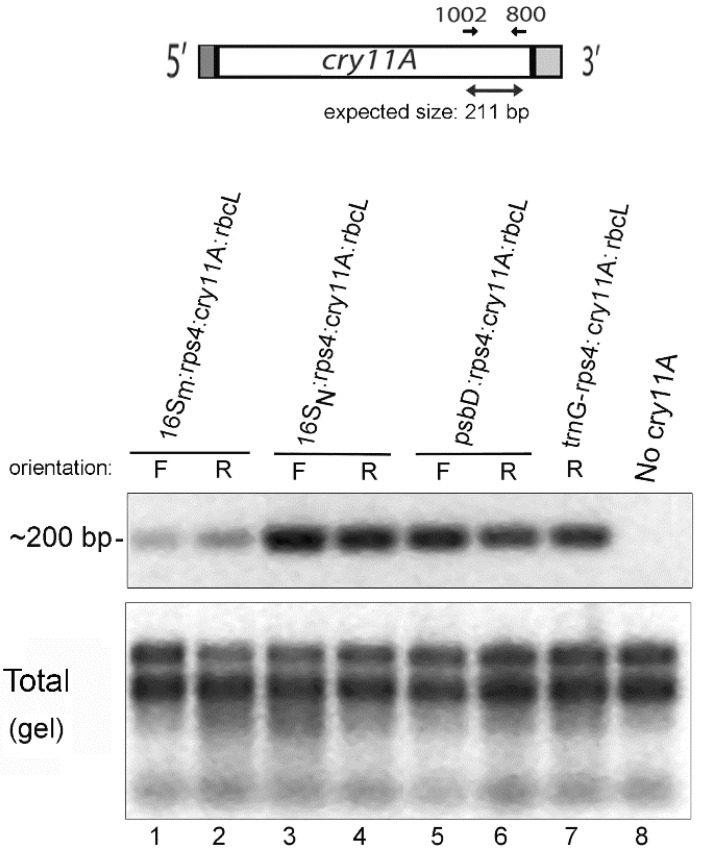
RT-PCR analysis of RNA expression levels for the same *rps4:cry11A:rbcL*-based transgenes with differing promoters that were analyzed in Figure 4. The diagram above the gels shows the location of the *cry11A* gene-specific primers used for RT-PCR. The upper gel photo is of the ethidium-DNA fluorescence gel of the RT-PCR products. Lanes 1–7 employed equal amounts of TNA/RNA from the same transformants used for protein analysis in Figure 4 plus a control transformant that lacked the *cry11A* transgene (lane 8). The lower gel image is of the total nucleic acid preparations, which were separated on agarose-ethidium bromide gel prior to DNase treatment and RT-PCR; only the RNA region of the gel is shown (the DNA migrated at the top). Both fluorescence gel images were inverted.

**Figure 6 microorganisms-10-01087-f006:**
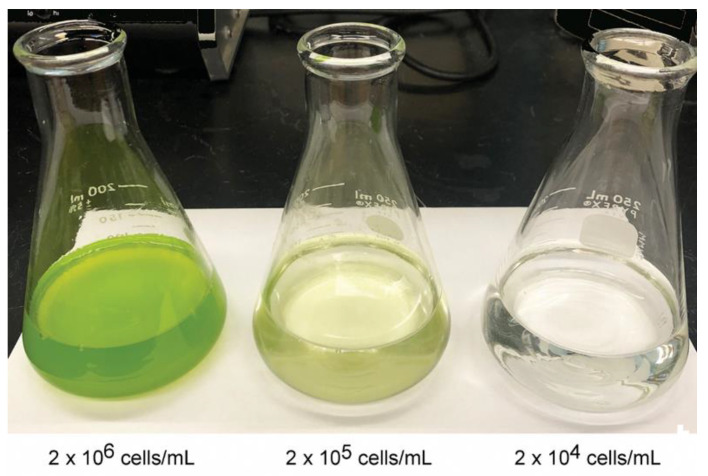
Representative cultures of *C. reinhardtii* at 2 × 10^6^, 2 × 10^5^, and 2 × 10^4^ cells/mL, respectively. A complemented transformant of CC-373 was grown to late log phase in TAP medium, centrifuged, resuspended in dH_2_0, and diluted with dH_2_0 to generate the 3 flasks shown in the figure.

**Figure 7 microorganisms-10-01087-f007:**
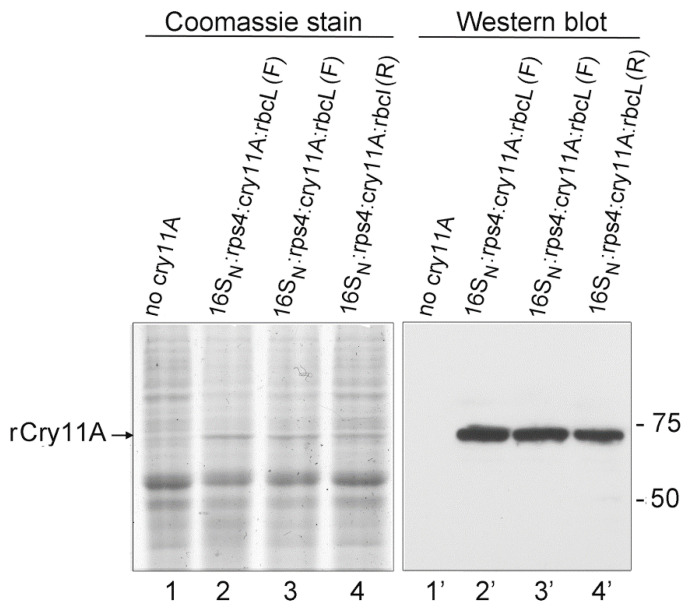
Stability of the *16S_N_:rps4:cry11A:rbcL* strains, and visualization of rCry11A on Coomassie stained gels of total protein. The two *16S_N_:rps4:cry11A:rbcL* (F) preparations (lanes 2–3) were made from the same transformant line 15 months apart. The total protein preparation from the *16S_N_:rps4:cry11A:rbcL* (R) transformant was made after carrying it for ~18 months. Duplicate 8% polyacrylamide gels were co-electrophoresed; one was stained with Coomassie blue (**left** panel, lanes 1–4) and the other was blotted and probed with the anti-Flag monoclonal antibody (**right** panel, lanes 1′–4′). The positions of two of the pre-stained size markers are indicated to the right of the blot, and the position of rCry11A on the left. The negative control lane (lanes 1 and 1′) was from a transformant that lacked the transgene.

**Table 1 microorganisms-10-01087-t001:** Bioassay with *Aedes aegypti* larvae and rCry11A (*16S_N_:rps4:cry11A:rbcL*) strains of *C. reinhardtii* at decreasing cell concentrations (algal cells do not grow during the assay).

	Strain	No. of Dead Larvae (Out of 12) ^a^			
Exper. 1		2 × 10^6^ cells/mL	1 × 10^6^ cells/mL	5 × 10^5^ cells/mL	2 × 10^5^ cells/mL
	*16S_N_:rps4:cry11A:rbcL* (R)	12/12	11/12	11/12	12/12
	Control (no Cry11A) ^b^	0/12	―	―	1/12

Exper. 2		1 × 10^6^ cells/mL	2 × 10^5^ cells/mL	8 × 10^4^ cells/mL	2 × 10^4^ cells/mL
	*16S_N_:rps4:cry11A:rbcL* (F)	10/12	11/12	10/12	10/12
	*16S_N_:rps4:cry11A:rbcL* (R)	10/12	11/12	7/12	7/12
	Control (no Cry11A)	0/12	0/12	1/12	1/12

^a^ The larvae were evaluated after 72 h. ^b^ The control strain was a transformant lacking the cry11A transgene.

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
