# Peer review of "Overcoming Poor Transgene Expression in the Wild-Type Chlamydomonas Chloroplast: Creation of Highly Mosquitocidal Strains of Chlamydomonas reinhardtii†"

_microorganisms, 2022, doi:10.3390/microorganisms10061087_

Round 1

Reviewer 1 Report

This study focuses on the development of Chlamydomonas reinhardtii strains capable of limiting the development of mosquitoes and black flies transmitting human parasitic or viral infections. Chlamydomonas and insect larvae can coexist in humid habitats offering the possibility of developing biological control tools such as the expression by Chlamydomonas of Bacillus thuringiensis Cry toxin which kills mosquito larvae. However, until now, the expression of transgenes in C. reinhardtii has remained a challenge due to a lack of knowledge of the sequences necessary for obtaining strong transcription and efficient translation in the chloroplasts of this organism.

The authors targeted the robust wild-type strain C. reinhardtii capable of growing in a wide variety of aquatic environments. Different constructs of transgenes expressing a Bacillus toxin have been constructed and integrated in C. reinhardtii.

Cry11A protein from Bacillus thuringiensis was chosen because of its most potent toxicity. The gene for Cry11A was codon-optimized for Chlamydomonas chloroplast and Ct-FLAG-tagged. Different constructs with different promoters were tested. The 5’ untranslated region of ribosomal protein rps4 was also added upstream the ORF of cry11A. The sequences of the added promoters and 5'UTR are provided, and the constructs of the transgenes have been carefully described in detail.

The first transgene that was tested in a wild-type background was basically a previous construct already tested in a benchmark strain of Chlamydomonas (Ind41_18). Toxin expression was analyzed by western blot and coomassie staining. However, Cry11A could only be expressed in the wild-type background at a much lower level than in the benchmark strain.

In a second construction, the strong 16S rRNA promoter and 5’ element from atpA gene were used, but without more success.

In another transgene, the ribosomal protein rps4 promoter and its 5’UTR were inserted but the expression Cry11A remained low.

In a last series of transgenes, different promoters (16S, tmG, psbD) were combined with the 5’UTR of rps4. The resulting ‘hybrid’ constructs showed in the wild-type background a 50% increase of cry11A expression compared to the benchmark strain Ind41_18. The expression reached 0.6% of the protein content.

Semi-quantitative RT-PCR experiments revealed that two transgenes were less efficiently transcribed whereas another one had normal RNA levels but was poorly translated.

The toxicity of strains carrying the most effective transgenes was tested in a bioassay in which Aedes aegypti larvae consumed algal cells expressing recombinant Cry11A toxin. Performed on a small number of larvae (12 per assay), the assay nevertheless clearly showed that the cells producing Cry11A were very toxic for Aedes aegypti larvae and could constitute an effective tool for the biological control of mosquito larvae.

Finally, the authors showed that the cry11A transgenes were stable in the strains over a period of about 2 years. The production of the rCry11A protein was also stable over time and its high level of expression now makes it possible to detect the protein by coomassie staining instead of western blot.

In the long term, this study proposes mosquito control solutions that are more respectful of the environment and more durable in humid environments.

The information I couldn't find was the origin of 16S, tmG-rps4 sequences. The authors should provide the name of the organism.

Reviewer 2 Report

In this MS, the authors explored overcoming poor transgene expression in chloroplasts of wild-type Chlamydomonas. There is a lot of content in this study, but there are still some problems in the manuscript that need to be improved, as follows:

  1. For the Latin name of the strain in the title, please write the full name.
    2. Please replace the first occurrence of the abbreviation in the abstract or the main text with its full name.
  2. Line79: Please put references at the end of the sentence.
  3. Please explain a little about the method of chlorophyll determination, for the current description is too simple.
    5. The results section describes too much background knowledge, so it is recommended to describe the results directly.
  4. Line54-56: the author suggested that transgenesC. reinhardtii as the host is used to control mosquito. Will the transgenes C. reinhardtii affect aquatic environment?

Round 2

Reviewer 2 Report

Thank for your careful revision.